# The Distinct Function and Localization of METTL3/METTL14 and METTL16 Enzymes in Cardiomyocytes

**DOI:** 10.3390/ijms21218139

**Published:** 2020-10-30

**Authors:** Orazio Angelo Arcidiacono, Jana Krejčí, Eva Bártová

**Affiliations:** 1Institute of Biophysics of the Czech Academy of Sciences, Královopolská 135, 61265 Brno, Czech Republic; o.arcidiacono@gmail.com (O.A.A.); krejci@ibp.cz (J.K.); 2Department of Experimental Biology, Faculty of Science, Masaryk University, Kamenice 753/5, 62500 Brno, Czech Republic

**Keywords:** cardiomyogenesis, epigenetics, epitranscriptomics, RNA methylation, METTL-like enzymes

## Abstract

It has become evident that epitranscriptome events, mediated by specific enzymes, regulate gene expression and, subsequently, cell differentiation processes. We show that methyltransferase-like proteins METTL3/METTL14 and N^6^-adenosine methylation (m6A) in RNAs are homogeneously distributed in embryonic hearts, and histone deacetylase (HDAC) inhibitors valproic acid and Trichostatin A (TSA) up-regulate METTL3/METTL14 proteins. The levels of METTL3 in mouse adult hearts, isolated from male and female animals, were lower in the aorta and pulmonary trunks when compared with atria, but METT14 was up-regulated in the aorta and pulmonary trunk, in comparison with ventriculi. Aging caused METTL3 down-regulation in aorta and atria in male animals. Western blot analysis in differentiated mouse embryonic stem cells (mESCs), containing 10–30 percent of cardiomyocytes, showed METTL3/METTL14 down-regulation, while the differentiation-induced increased level of METTL16 was observed in both wild type (wt) and HDAC1 depleted (dn) cells. In parallel, experimental differentiation in especially HDAC1 wild type cells was accompanied by depletion of m6A in RNA. Immunofluorescence analysis of individual cells revealed the highest density of METTL3/METTL14 in α-actinin positive cardiomyocytes when compared with the other cells in the culture undergoing differentiation. In both wt and HDAC1 dn cells, the amount of METTL16 was also up-regulated in cardiomyocytes when compared to co-cultivated cells. Together, we showed that distinct anatomical regions of the mouse adult hearts are characterized by different levels of METTL3 and METTL14 proteins, which are changed during aging. Experimental cell differentiation was also accompanied by changes in METTL-like proteins and m6A in RNA; in particular, levels and distribution patterns of METTL3/METTL14 proteins were different from the same parameters studied in the case of the METTL16 protein.

## 1. Introduction

Accurate gene expression, cell fate programs, or cell stress response require harmonious interplay between all regulatory levels, including epigenetic and epitranscriptomic mechanisms. For instance, a significant regulatory role is ascribed to 5-methylcytidine (m^5^C), 5-hydroxymethylcytidine (hm^5^C), N4-acetylcytidine (ac^4^C), and N^6^-methyladenosine (m^6^A) in distinct types of RNAs. Among the over 100 chemically different modifications in coding and non-coding RNA discovered so far [1], the N6-methyl adenosine (m6A) is the most abundant and influential modification of mRNA [2], also found in non-coding RNAs (ncRNAs) and long non-coding RNAs (lncRNAs) [3,4]. The orchestrated interplay among writers, erasers, and readers drives the dynamics and outcomes of the m6A modification of RNAs. The appearance of the methyl group in RNAs is a co-transcriptional event, occurring in nuclear speckles, and involves the recruitment of the METTL3 protein, characterized by the catalytic activity, in pre-mRNA. This regulatory process mostly appears in intron regions [5,6,7,8]. The deposition of the methyl group is guaranteed by a multiprotein complex composed of the heterodimer methyltransferase-like 3/methyltransferase-like 14 (METTL3/METTL14) [9,10] and the m6A-METTL-associated complex: WTAP, KIAA1429, RBM15, HAKAI, and ZC3H13 (MACOM) [10,11,12,13] or to the active monomer methyltransferase-like 16 (METTL16), as recently documented [14,15]. A significant eraser of m^6^A in mRNA is fat mass and obesity-associated protein (FTO) [16] and its homolog, alkB homolog 5 (ALKBH5) [17]. These proteins can oxidatively reverse N^6^-adenosine methylation in nuclear RNA [16]. It is well-known that FTO belongs to the alkB subfamily of dioxygenases, dependent on FeII/α-ketoglutarate [17,18,19,20]. Functionally, FTO oxidizes m6A in RNAs to N6-hydroxymethyladenosine (hm6A) and N6-formyladenosine (f6A) [21]. This process seems to be similar to the oxidation of 5-methylcytosine (5mC) in genomic DNA to 5-hydroxymethylcytosine (5hmC) and finally to 5-formyl-cytosine (5fC). However, this process in DNA is mediated by ten-eleven translocation (TET) enzymes [22,23,24].

The m6A modification influences almost all metabolic processes of mRNA, such as mRNA nuclear export [17,25], RNA folding (m6A switch) [26], mRNA maturation [26,27,28,29], or mRNA decay. From the view of ESC biology, knockdown of METTL3 and METTL14 increased the expression of their mRNA targets [10,30,31]. Moreover, recently, the function of m6A in RNA has been also linked to DNA damage response where METTL16 methylates the small RNAs in the vicinity of the DNA lesions [32]. The latest studies also point out the pivotal role of m6A RNA in shaping the cell state during mouse embryonic stem cell (mESC) differentiation. For instance, the m6A in RNA is required for the proper transition from naïve to primed state, allowing the replacement of pluripotency transcriptome to establish a new cell identity [30,33,34]. Specifically, it has been found that the transcript of the pluripotency factors, such as NANOG, OCT4, SOX2, and KLF4, are marked by m6A [34,35,36]. Interestingly, the partial depletion of METTL3 or METTL14 by short hairpin RNAs reduced self-renewal in mESCs [30], but the knock-out of METTL3 (by the CRISP-Cas9 system) impeded the mESC differentiation [34]. These conflicting data reflect the specific effect of m6A modification in distinct RNAs, considering that in naïve ESCs (derived from the Inner Cell Mass of preimplantation embryos), the METTL3 depletion (associated with m6A RNA decreasing) induces the so-called hypernaïve pluripotent state. However, in the primed Epiblast Stem Cells, characterized by a high transcription activity, the loss of METTL3 accelerates the differentiation process [33]. Along with these recent discoveries on epitranscriptome changes, also the role of epigenetic regulations, including histone post-translational modifications, has been proven to be fundamental in stem cell biology and development. Both epigenetic and epitranscriptome changes affect cardiomyogenesis. For instance, the histone deacetylase 1 (HDAC1) depletion in a mice model led to the death of embryos at day 10.5 (E 10.5). Moreover, proliferation defects can be observed in HDAC1 depleted (dn) mouse embryonic stem cells (mESCs) with an increased expression of cyclin-dependent kinase inhibitors p21 and p27 [37,38,39]. The depletion of both HDAC1 and HDAC2 is responsible for cardiac arrhythmias, dilated cardiomyopathy, and upregulation of contractile proteins and calcium channels in the heart [39]. Using an in vitro mESC differentiation model into cardiomyocytes [40], in our previous study, we pointed out the different beating duration between cardiomyocytes established from HDAC1 wild type mESCs and HDAC1 (dn) mES cells [41].

Based on the results mentioned above, we addressed the question whether METTL-like proteins (METTL3/METTL14 and METTL16) work in a synergic way during cardiomyogenesis, induced in wild type (wt) and HDAC1 (dn) mouse embryonic stem cells. Moreover, we hypothesized that epitranscriptomic features could be changed during cardiomyogenesis. Moreover, we tried to reveal epigenomic/epitranscriptomic distinctions between young and old mouse hearts.

## 2. Results

### 2.1. METTL3/METTL14 Proteins are Homogeneously Distributed in Embryonic Hearts’ Cryosections and the Highest Density of METTL16 is in the Right Atrium of Embryonic Hearts

We studied the levels of METTL3, METTL14, METTL16 enzymes, and m6A RNA in the following regions of embryonic hearts (stage e15): right atrium (RA), left atrium (LA), aorta (AO), right ventriculus (RV) and left ventriculus (LV) or intraventricular septum (IVS). In the middle section of e15 embryonic hearts, by immunofluorescence combined with tile scaling, we observed relatively identical distribution of METTL3, METTL14 proteins, and m6A RNAs (non-significant changes appeared, as shown in Table 1), while especially RA was characterized by the significantly higher density of the METTL16 protein in comparison to LA, IVS, RV, LV, and AO. For explanation, according to the measured fluorescence intensity (FI), the level of the METTL16 protein was three times higher in the right atrium, compared to the aorta and pulmonary trunk (Figure 1A(a–d) and Figure 1B(a–d); Table 1). Other changes were not so pronounced, except a high level of METTL16 in LA; moreover, this anatomical region was additionally characterized by the highest level of m6A in RNA when it was compared with the aorta (Figure 1B(d), Table 1).

### 2.2. Epigenomic and Epitranscriptomic Features in Explanted Young and Old Mice Adult Hearts

In mouse hearts isolated from young and old animals (Figure 2A), we analyzed the level of METTL3, METTL14 proteins, and histone markers in ventricular parts, atria, and in the region covering aorta and pulmonary trunk (Figure 2A–D). Samples were isolated from young (2 months) and old (27 months) mice of both males and females. We observed METTL3-positivity in all tested anatomical regions; the lower density of METTL3 was in aortas and pulmonary trunks in comparison to atria (Figure 2B,D). METTL3 down-regulation was found in aorta/pulmonary trunks and atria in aging males (see asterisks in Figure 2D). In comparison to ventriculi, the level of METTL14 was higher in the aorta/pulmonary trunk or atria of young animals when compared to old individuals (Figure 2B,D). Importantly, HDAC1 wt and HDAC1 dn mESCs were characterized by higher levels of METTL3 and METTL14 proteins in comparison with and adult heart tissues of male and female origin (Figure 2B). METTL16 was under the detection limit in adult mouse hearts in comparison with embryonic states e15 and e13 (Figure 3C); thus, we did not show these Western blot results in Figure 2B. Together, we documented that in various anatomical regions of mouse hearts, the levels and density profiles of METTL3/METTL14 are distinct from that studied in the case of the METTL16 protein.

Moreover, in comparison to males, the level of HDAC1 was higher in the atria of young and old female animals. Compared to young males, HDAC3 was up-regulated in aortas/pulmonary trunks and atria of old males, additionally characterized by a higher level of HDAC3 when it was related to HDAC3 profile in female animals (Figure 2C,D). As a summary, we can conclude that cardiomyogenesis is accompanied by changes in the epigenome and epitranscriptome. This is supported by the fact that the levels of epigenetic markers H3K4 di-methylation (H3K4me2) and H3K9 tri-methylation (H3K9me3) are distinct in various anatomical regions of adult mouse hearts. For example, Western blot data, normalized to the total protein levels, showed a high density of H3K4me2 in atria of young and old females when compared to male animals (Figure 2C,D). H3K9 tri-methylation was lower, especially in the ventriculi, in comparison with aortas/pulmonary trunks (Figure 2C,D). Moreover, the level of HDAC1 and both H3K4me2, H3K9me3 was higher in wt mouse ECSs when compared with adult mouse hearts (Figure 2C).

### 2.3. Levels of METTL3/METTL14/METTL16 Proteins in Mouse Embryonic Hearts (e15) Treated with Inhibitors of HDACs (HDACi)

By Western blots, we analyzed the levels of METTL3, METTL14, and METTL16 proteins in non-treated, TSA-, valproic acid (VPA)-, and suberoylanilide hydroxamic acid (SAHA)-treated explanted hearts from mouse embryos at stage e15. We quantified Western blot fragments of the expected protein’s weight, although the membranes show multiple bands, probably due to different protein isoforms. We observed that HDAC inhibitors TSA and VPA increased the levels of all proteins studied. Statistically significant increase in METTL3 we found in e15 hearts treated with VPA when compared with non-treated tissue (Figure 3A). A considerable increase was also found in the case of the METTL14 protein studied in explanted hearts from e15 embryo and treated for 3 h with TSA (Figure 3B). Moreover, the level of METTL14 was reduced in SAHA-treated embryonic hearts, similarly, SAHA reduced the METTL16 level (Figure 3B,C). VPA-induced change in the level of METTL16 was not statistically significant (Figure 3C). Interestingly, adult hearts were characterized by subtle levels of METTL3 and METTL14 proteins, and there was an absence of the METTL16 protein in comparison to explanted hearts isolated from e13 embryos (Figure 3A–C). However, e13 hearts were characterized by a lower level of METTL3/METTL14 in comparison to the e15 stage (Figure 3A,B). The effect of HDAC inhibitors were also analyzed with the view marker, characterized for cardiac differentiation, α-actinin (Figure 3D). Moreover, Siciliano et al. [42] showed that troponin I is a marker of myocardial damage. In this case, we show that treatment by all studied HDACi increased the level of α-actinin, but only valproic acid (VPA) up-regulated troponin, the level of which was significantly higher in adults’ hearts in comparison to the hearts isolated from e13 and e15 embryos (Figure 3D).

### 2.4. Decreasing Levels of METTL3 and METTL14 Proteins Were Observed in Mouse ESCs Undergoing Differentiation while the METTL16 Protein Was Up-Regulated

Western blots showed the decreasing levels of METTL3 and METTL14 proteins in wt and HDAC1 dn mouse ESCs undergoing experimentally-induced cell differentiation, of which efficiency was monitored 10, 15, and 20 days after experimental initiation of differentiation processes (Figure 4A,B). Conversely, in comparison to non-differentiated mESCs, METTL16 was up-regulated, especially in days 15 and 20 of experimentally-induced differentiation (Figure 4C). We observed the described trends in both control wt mESCs and HDAC1 (dn) mESCs (Figure 4A–C).

In the differentiation pathway, studied in HDAC1 wt and HDAC1 dn cells, we additionally analyzed the level of total m6A in RNAs, and we observed a depletion of m6A in RNAs in wt cells differentiated 10, 15, and 20 days into cell population containing cardiomyocytes, in comparison to non-differentiated cells (day 0) (Figure 4D, squares). A slight depletion, but not statistically significant, of m6A in RNA, was also in HDAC1 dn cells, analyzed at day 10 of differentiation (Figure 4D). In comparison to wt mESCs, HDAC1 dn mESCs, at day 15 and 20 of differentiation, were characterized by a significantly higher level of m6A in RNAs (Figure 4D, asterisks). According to these trends, it seems likely that m6A in RNA in especially wt mESCs is linked to the function of the METTL3 and METTL14 enzymes, the level of which level decreased during differentiation.

### 2.5. METTL3, METTL14 and METTL 16 in α-actinin Positive Cells Were Higher than in the Cells Absent of α-Actinin

Analysis of immunofluorescence data showed upregulation of especially METTL3 and METTL14 in α-actinin positive cells that are considered to be cardiomyocytes (Figure 5A–D and Figure 6A,B). For explanation, the levels of METTL3/METTL14 were different when we analyzed whole-cell populations containing up to 30% of cardiomyocytes by Western blots on one side and α-actinin positive cardiomyocytes and non-cardio cells in the same cell population by immunofluorescence (IF) on another side. In this case, IF enables analysis on an individual cellular level; thus, it was possible to distinguish α-actinin positive and α-actinin negative cells (Figure 6A–C). Some cells, absent of α-actinin, were characterized by METTL3 and METTL14 low level or depletion (Figure 5A,B, and Figure 6A,B). In such a heterogeneously differentiated cell population, Western blots showed METTL3/METTL14 down-regulation (Figure 4A,B).

In addition, we observed that the fluorescence intensity of the METTL16 protein in α-actinin positive cells increased (Figure 5C and Figure 6C), while in the non-cardio cell population, the level of the METTL16 protein was nearly identical with the control values (Figure 5C). However, Western blots, showing the results in the whole cell population, revelated an upregulation of METTL16 in the heterogeneously differentiated cell population (Figure 4C). Note: in the studied cell culture with cardiomyocytes, only 10–30 % of cells are α-actinin positive [43].

### 2.6. Distribution Pattern of METTL-Like Proteins and m6A RNAs in Cell Nuclei and the Cytoplasm of mESCs

Immunofluorescence analysis showed a high level of m6A RNAs in the cytoplasm of both HDAC1 wt and HDAC1 dn mESCs, while the subtle level of this post-transcription modification we observed inside the cell nuclei. On the other hand, we found the high density of METTL-like proteins (METTL3/METTL14 and METTL16) mainly in the nucleoplasm, and METTL14 also appeared in the cytoplasmic fraction (Figure 7A–C). Importantly, in mESCs, the METTL16 protein was abundant in the whole cell nucleus, with the highest density in the compartment of nucleoli. Moreover, we observed similar nucleolar localization of METTL16 in human cell lines, A549, U2OS, and HeLa.

## 3. Discussion

It is well-known that the crucial role of epigenetic regulation of mESC pluripotency on one side and specific differentiation on another side is promising from the view of medical applications. Similarly, an understating of epigenomic/epitranscriptomic features of the heart is essential for efficient heart failure therapy.

### 3.1. Changes in Epigenome Influence the Characteristics of the Epitranscriptome Over the Differentiation Process

In the present study, we addressed this aspect from the viewpoint of knowledge on epitranscriptomic regulation, uncovering the dynamics and functional properties of METTL-like enzymes and their role during mESC differentiation, particularly into cardiomyocytes. We have also tried to find a link between m6A RNAs’ profile and histone posttranslational modifications in heart tissue (Figure 2A–C). For example, Wang et al. [44] showed that m6A regulates histone modification via the destabilization of transcripts encoding histone-modifying enzymes. Moreover, Huang et al. [45] documented that m6A modification is dense in close proximity to H3K36-trimethylated chromatin. When in mouse embryonic stem cells, H3K36me3 is decreased, and m6A is also reduced in many transcripts. We observed that the depletion of METTL3 in aortas and pulmonary trunks is accompanied by high levels of H3K4me2 and H3K9me3 (Figure 2B–D). In this case, we show a link between epitranscriptomic features and histone signature.

### 3.2. METTL3/METTL14 and METTL16 Proteins are Up-Regulated in α-Actinin Positive Cardiomyocytes

We used the embryoid bodies (EBs) as a study model, which enables us to induce differentiation toward the three germ lineages and gives rise to a multicellular arrangement, including the cardiomyocytes. By the use of this experimental model, we tried to better understand the epigenetic and epitranscriptomic background of the HDAC1 depleted ES cells and their wild-type counterpart. Really interesting, in both cell lines, were the profiles of METTL3 METTL14 and METTL 16 proteins in differentiated cell-populations. In fact, changes in METTL3 and METTL14 levels were distinct from the levels of the METTL16 protein when studied in whole-cell populations (Figure 4A–C). However, the differentiation into cardiomyocytes requires a specific epitranscriptomic regulation, different from the rest of the differentiated-cells in the population consisting of only 30% of α-actinin positive cardiomyocytes, requiring an increase in METTL3, METTL14, and METTL16 proteins (Figure 5A–D). 

Importantly, the decreasing level of m6A in RNAs during the differentiation of HDAC1 wt mESCs was consistent with the decreasing profile of METTL3 and METTL14 proteins. This was also the same in HDAC1 dn differentiated cell population (compare dd0 with dd10), but not statistically significant (Figure 5D). In the case of HDAC1 dn cells, the m6A RNA level was different from those observed in HDAC1 wt cells, especially when we compared day 15th and 20th of differentiation (Figure 5D). Again, this observation confirms that the depletion of HDAC1 (changes in epigenome) affects the epitranscriptomic profile during the differentiation process.

Here, we must additionally admit that the role of the METTL16 protein remains still elusive. We only know that subcellular accumulation of METTL16 in the nucleolar compartment differs substantially from those of METTL3 and METTL14 proteins; thus, one can believe that METTL16 protein, conversely to METTL3/METTL14, regulates m6A in distinct RNAs [46,47]. Thus, it must be further specified whether the multiprotein complex composed of the heterodimer METTL3/METTL14 [9,10] works in an identical or distinct way to METTL16. Here, in both studied cell types (HDAC1 wt and HDAC1 dn), we found that the METTL16 shows an opposite, increasing trend in the protein levels when compared to METTL3/METTL14 proteins, studied in the whole differentiated population (Figure 4A–C). Other studies show that the METTL16 protein is considered an important player during mouse embryonic development. In this case METTL16 regulates the MATA2 mRNA level [13,46,48]. Considering that the methyltransferase activity of METTL16 is also directed towards non-coding RNAs or rRNA, and METTL3/METTL14 mediate m6A in mRNA [14], our data suggest a distinct role of m6A in non-coding RNAs and mRNA over the cell differentiation. However, further studies are required to fully elucidate the biological function of m6A in distinct RNAs contributing to cardiomyogenesis and development.

### 3.3. Homeostasis in Cardiomyocytes and Responses to Hypertrophic Stimuli

Recent studies also investigated the role of m6A RNA modification in the heart, highlighting its importance in cardiac homeostasis and diseases. Dorn et al., using a METTL3-overexpressing mice model, documented dose-dependent cardiac hypertrophy, while knockdown of METTL3 on neonatal rat cardiomyocytes (NRCM) blocked the hypertrophy after hypertrophic stimuli and METTL3 knock out mice showed sign of heart failure [49]. Conversely, Kmietczyk et al. recorded the mitigation of hypertrophic growth in METTL3-overexpressing mice after transverse aortic constriction (TAC) and increased cell size in Mettl3-knockdown NRCM after hypertrophic induction [50]. Although showing contradictory results, both studies point out the relevance of the Mettl 3 gene in the homeostasis of cardiomyocytes. The m6A RNA sequencing (meRIP-seq) revealed that approximately one-quarter of transcripts in healthy mouse and the human heart is m6A methylated, mostly in 5′UTR, 3′UTR and coding sequence [51], while an increased m6A RNA level in failing human myocardium [50,52] or in cultured rat neonatal cardiomyocytes was found in response to hypertrophic stimuli [49]. Gene Ontology term analyses showed that in human failing heart samples or a mouse model of aortic failure, including TAC, the m6A-containing transcript is enriched for transcripts encoding transcriptional regulators and heart development genes [50,51]. Interestingly, the transcripts differentially m6A methylated (hypomethylated or hypermethylated) were higher than differentially expressed genes, mainly linked to the structural cardiac plasticity pathway [51]. In the human heart’s failing tissue, for example, the level of the Calmodulin 1 protein is reduced while its mRNA level is unchanged, but the m6A RNA level decreased [51]. Considering that the change in m6A methylation profile mainly occurs in the transcripts involved in cardiac signaling and metabolic processes, as a consequence, changes in the expression of these regulatory targets by the m6A methylation are turned into downstream effects that influence the cardiac phenotype in response to cardiac stress [50,51]. Together, these studies point out the remarkable role of epitranscriptomic regulations in heart development, homeostasis, as well as heart failure. We put these results into the context of the status of markers, specific for cardiomyogenesis and myocardial damage, including α-actinin and troponin (Figure 3D). We observed that HDAC inhibitors likely potentiate cardiomyogenesis because the α-actinin level was enhanced in e15 hearts treated with HDACi (Figure 3D). Surprisingly, only valproic acid (VPA), but not TSA or SAHA (clinically used HDACi), did not induce up-regulation of troponin; thus, damage of cardiomyocytes seems to be induced only by VPA. Interestingly, adult hearts were characterized by a more pronounced level of troponin in comparison to embryonic ones (Figure 3D), which show a higher predisposition of adult heart tissue to damage.

According to the present knowledge, it is well-known that the functions of HDACs and METTL-like enzymes are essential for cardiac functions [49,53,54]. Moreover, as mentioned above, HDAC inhibitors can affect cardiac functions, for example, affecting hypertrophy, autophagy processes, arrhythmogenesis, fibrosis, and heart failure [39,54,55,56,57]. In this context, our observation regarding the effects of HDACi treatment of mouse embryonic hearts at stage E15 (Figure 3), showing that especially TSA and VPA treatment increased the level of METTL3/METTL14 proteins and α-actinin or troponin is really interesting. Thus, taken together, our data show a link between HDAC1 epigenetic regulation, METTLs enzyme function, the level of m6A in RNAs, and the status of myocardial damage marker troponin.

### 3.4. Conclusions and Future Directions

Here, we confirm functional crosstalk between histone signature and RNA methylation on N^6^-adenosine, as also suggested by Huang et al. [45] in mouse embryonic stem cells. We want to suggest that besides differentiation processes, the effect of aging and gender should also be considered from the view of epigenetic/epitranscriptomic regulations. For instance, it is well-known that in humans, men suffer from cardiovascular diseases (CVD) more frequently than women whose cardiovascular system is protected by female hormones [58]. In this regard, changes in histone signature, as well as post-transcription modifications of RNAs, should be considered from many angles, including cell differentiation, embryonic development, and aging.

## 4. Materials and Methods

### 4.1. Mouse ESCs Cultivation and Differentiation

Epigenetic aspects and the function of HDACs were studied in mESCs: wild-type (wt) mESCs, D3 line (American Tissue Culture Collection, ATCC, CRL-11632) and mESCs that were deficient in HDAC1 (HDAC1 depleted (dn) mESCs, a generous gift from Max F. Peruz Laboratories in Vienna, Austria). Mouse ESCs were cultivated on 0.2% gelatin-coated Petri dishes (valid for wt-cells) or Matrigel (#354277, Thermo Fischer Scientific, Rockford, IL, USA)-coated plastic dishes (valid for HDAC1dn-cells). Mouse ESCs were grown in Dulbecco’s Modified Eagle Medium (DMEM, Sigma-Aldrich, Prague, Czech Republic) supplemented with penicillin and streptomycin, 0.1 mM non-essential amino acids, 1 ng/mL mouse leukemia inhibitory factor (LIF), 100 µM mono-thioglycerol, and 15% fetal bovine serum (FBS). Differentiation protocols showing mESCs stimulation into cardiomyocytes via EBs were described in Arcidiacono et al. [41]. Cardio differentiation was initiated by seeding 400 cells per 30 µL into ES culture media without the addition of the LIF factor using the “hanging drop” method. On the 3rd day of differentiation (dd3), the increased embryonic bodies (EBs) were plated onto non-adhesive (bacteriological) plastic dishes; on dd6, EBs were transferred to gelatin-coated culture dishes with DMEM/F12 (1:1) (#11320-033, Gibco, Paisley, UK) supplemented with insulin, transferrin, and selenium (ITS-100x, #41400-045, Gibco) (DMEM/F12-ITS). The serum-free DMEM/F12-ITS culture medium was changed every two days. Differentiated cells into cardiomyocytes were harvested in interval days of differentiation 10, 15, 20, and 25 (dd10, dd15, dd20, dd25) for the next analyses.

Human A549 (ATTC, CCL-185), U2OS (ATCC, HTB-96), and HeLa cells (ATCC, CCL-2) were cultivated in DMEM, supplemented with 10% of fetal calf serum. All cell lines were maintained at 37 °C in a humidified atmosphere containing 5% CO_2_.

### 4.2. Heart Sectioning and Tissue Fixation

For experiments with animals, we used C57Bl6 mice purchased from the Breeding Facility of the Medical Faculty, Masaryk University, Brno, Czech Republic. Mice were kept and fertilized under standard conditions, and their use followed the European Community Guidelines of accepted principles for the use of experimental animals. Isolation of mouse hearts was performed according to an experimental protocol confirmed by the National and Institutional Ethics Committee (protocol No.: 48/2016, valid: 22/0600/2016-31/12/2021). Mice were sacrificed by cervical dislocation. Embryonic hearts were explanted from embryos on 15.5th day of embryonic development (e15) and were treated with HDAC inhibitors (200 nM Trichostatin A (TSA), 16 µM suberoylanilide hydroxamic acid (SAHA), and 15 mM valproic acid (VPA)) diluted in DMEM with 10% FBS for 3 h at 37 °C. Adult hearts were obtained from animals age 2 and 27 months; experimental groups of young and old mice were brothers and sisters. Explanted embryonic and adult hearts for Western blot analysis were disaggregated in 1% SDS lysis buffer. For immunoprocessing, the tissues were fixed in 4% paraformaldehyde, washed in PBS, incubated in 30% sucrose, embedded into the tissue-freezing medium (OCT embedding matrix, Leica Microsystems, Mannheim, Germany), and stored at −20 °C before sectioning. The tissues were sectioned using a Leica cryo-microtome (Leica CM1800, Leica, Germany).

### 4.3. Immunofluorescence on Mouse Embryonic Cells and Heart Sections and Confocal Microscopy

For immunofluorescence, we used hearts isolated from the e15 embryo and mESCs differentiated in monolayer for 20 days (dd20). Cryo-sections of embryonal e15 hearts were washed in PBS and post-fixed by 4% formaldehyde for 5 min. In vitro differentiated cells were fixed by 4% formaldehyde for 20 min. In both, the sample permeabilization was performed in 1% Triton X-100 and 0.1% saponin in 0.1% Triton-X100 (Sigma-Aldrich, Prague, Czech Republic) dissolved in PBS. As a blocking solution, we used 4% bovine serum albumin (BSA) dissolved in PBS. Incubation took place for 1 h at room temperature. Primary antibodies, used for immunofluorescence, were the following: anti-METTL3 (A8370; AB Clonal); anti-METTL14 (A8530; AB Clonal); anti-METTL16 (HPA 020352; Sigma-Aldrich); anti-α-actinin (sarcomeric) (A7811; Sigma-Aldrich); anti-N6-methyladenosine (202111; Synaptic System; SYSY, Goettingen, Germany).

After overnight incubation with primary antibodies at 4 ^°^C, samples were washed in PBS three times for 5 min each and incubated with the appropriate secondary antibodies at room temperature for 2 h. Secondary antibodies used were the following: Alexa Fluor 488-conjugated donkey anti-mouse IgG (H+L) (A21202), Alexa Fluor 594-conjugated donkey anti-rabbit IgG (H+L) (A21207, both Life Technologies, Eugene, OR, USA). As negative controls, samples were incubated without primary antibody or with rabbit control IgG (H+L) (ab46540-1, Abcam, Cambridge, UK) before incubation with secondary antibody. Primary antibodies were used at a dilution of 1:100 and secondary antibodies at a dilution of 1:200 in PBS containing 1% BSA. Nuclei (DNA content) were counterstained with 4’,6-diamidino-2-phenylindole (DAPI; Sigma-Aldrich), and as a mounting medium, we used Vectashield (Vector Laboratories, Burlingame, CA, USA).

For image analysis, we used a Leica SP5–X laser scanning confocal microscope Leica Microsystems, Germany). Image acquisition was performed by LAS AF software (Leica microscopy, Mannheim, Germany) at the following setting: 1024 × 1024 pixels, 200 Hz, zoom 1, oil objective (HCX PL APO, 63×, NA = 1.4, 1024 × 1024 pixels at objective 20× (HCX PL APO lambda blue (20.0 × 0.7 IMM UV, Leica Microsystems, Mannheim, Germany). Fluorochromes used were the following: Alexa Fluor^®^594 (max. emission/max. excitation was 590/617 nm), and Alexa Fluor^®^488 (max. emission/max excitation was 495/512 nm), and DAPI (max. emission/max. excitation was 358/461 nm). Contours of heart sections were inspected under transmitted light.

We monitored 200–300 cell nuclei of mESCs, and 20 cryo-sections of mouse hearts, and we performed our analyses with the use of 2–3 biological replicates, containing 2–3 technical replicates. The images from confocal microscopes were acquired in an 8-bit setting, corresponding to a display range of 256 gray levels. LAS AF software (Leica microscopy, Mannheim, Germany) was used for image acquisition and analysis of fluorescence intensity.

### 4.4. Tile-Scanning

Cryo-sections (10 µm) of whole embryonic hearts (at stage e15) were stained with primary antibody: anti-METTL3 (A8370; AB Clonal); anti-METTL14 (A8530; AB Clonal); anti-METTL16 (HPA 020352; Sigma-Aldrich); anti-α-actinin (A7811; Sigma-Aldrich); anti-N6-methyladenosine (202111; Synaptic System), to visualize the distribution pattern of METTLs or m6A in RNAs. For image acquisition, we used the tile-scanning mode and the following objective: HCX PL APO lambda blue 20.0 × 0.7 IMM UV (Leica Microsystems) or HCX PL APO, 63×, NA = 1.4. Scanning was performed at a resolution of 1024 × 1024 pixels, 200 Hz, and for image reconstruction, we used the auto-stitched tile-scanning mode involving the smooth-scanning mode with a slow/fine speed accuracy (set in the Leica LAS AF software connected to the Leica SP5 microscope).

### 4.5. Western Blotting

For analyses, we used the following primary antibodies: anti-METTL3 (A8370; AB Clonal); anti-METTL14 (A8530; AB Clonal); anti-HDAC1 (sc-7872; Santa Cruz Biotechnology, Dallas, TX, USA); anti-HDAC3 (SAB1404635, Prague, Sigma Aldrich); anti-H3K4me2 (07–030, Upstate, St. Louis, MO, USA); anti-H3K9me3 (ab8898, Abcam, Cambridge, UK); anti-α-actinin (A7811; Sigma-Aldrich, Prague, Czech Republic); and anti-troponin (ab47003, Abcam, Cambridge, UK). The following secondary antibodies were used for Western blot analyses: anti-rabbit IgG (A4914, Sigma Aldrich, Prague, Czech Republic) and anti-mouse IgG (A9044, Sigma Aldrich, Prague, Czech Republic). Protein concentration was measured by µQuant^TM^ microplate spectrophotometer (BioTek, see at https://www.artisantg.com/Scientific/71777-1/Bio-Tek-uQuant-Universal-Microplate-Spectrophoto-meter). Gels were stained by amido-black 10 B. The density of Western blot fragments was measured, and data were normalized to the level of total proteins. All blots were imaged on Amersham Imager 680. The final quantification of protein levels was performed using Image Studio Ver.5.2.

### 4.6. Analysis of the Total m6A RNA Methylation

Total RNA was isolated by the use of GenElute Total RNA Purification Kit (#RNB100; Sigma-Aldrich) from cells differentiated into cardiomyocytes. The m6A RNA status was detected directly by the use of the total RNA Methylation Quantification Kit (Colorimetric) (#ab185912; Abcam) according to the manufacturer’s instructions. In detail, the RNA binds to assay wells, the washing was by washing buffer (a component of kit), and then the capture antibody was added. After incubation, wells were washed, and the detection antibody was used. After that, the enhancer solution was added. Quantification was done after the adding of color-developing solution. The absorbance was measured by a µQuant spectrometer, and data were summarized in Excel software.

### 4.7. Statistical Analysis

A nonparametric Mann–Whitney U test (STATISTICA software; © SAS Institute Inc., Marlow, Buckinghamshire, UK) was used for data analysis. As the fist, we tested the null hypothesis that is applied for X and Y values, randomly selected from two experimental units. For the analysis, we used the following calculator (https://www.socscistatistics.com/tests/mannwhitney/default2.aspx). For selected data, we also applied the unpaired and also the paired Student’s t-test.

## Figures and Tables

**Figure 1 ijms-21-08139-f001:**
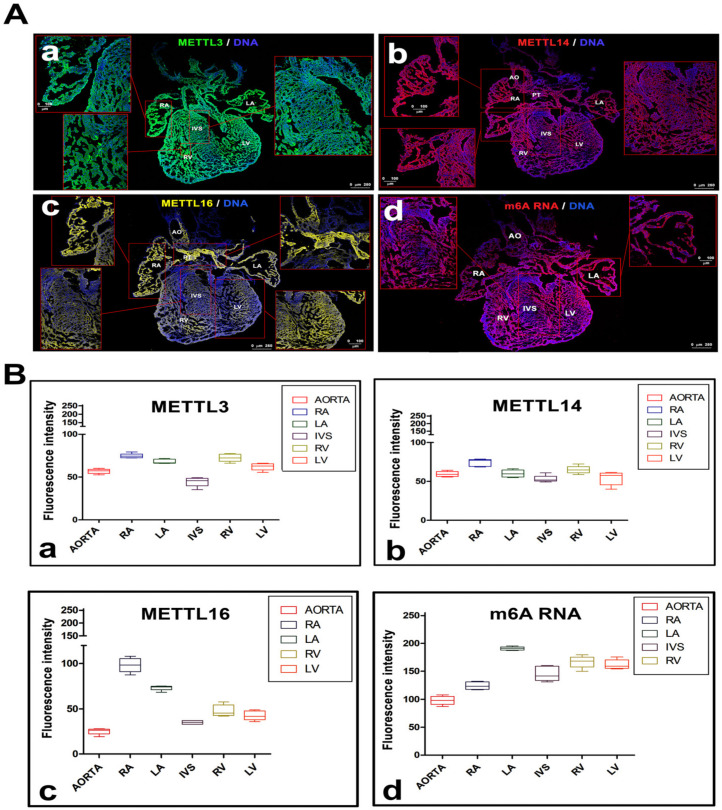
The distribution pattern of methyltransferase-like METTL3, METTL14, METTL16 proteins, and N^6^-adenosine methylation (m6A) in RNAs in mouse embryonic hearts, isolated at stage e15. (**A**) Immunofluorescence analysis of (**a**) METTL3, (**b**) METTL14, (**c**) METTL16 proteins and (**d**) m6A RNAs was done in the right atrium (RA), left atrium (LA), aorta (AO), right ventriculus (RV) and left ventriculus (LV) or intraventricular septum (IVS). Scale bars in the main panels Aa-d show 100 µm. (**B**) Immunofluorescence (IF) in panels **A**(**a**–**d**) was quantified by LAS AF software (Leica microscopy, Mannheim, Germany). Data are shown for (**a**) METTL3, (**b**) METTL14, (**c**) METTL16 proteins, and (**d**) m6A RNAs. IF was studied in anatomical regions, described in panels **A**(**a**–**d**). Experiments were repeated twice, and for each experimental event, 20 cryo-sections of mouse embryonic hearts were acquired by tile scanning microscopy. Panels **A**(**a**–**d**) show representative examples of protein distribution that were quantified in panels Ba-d. Quantification was done in the following way: in each individual measurement, FI of background in the given channel was subtracted from FI acquired for the given protein.

**Figure 2 ijms-21-08139-f002:**
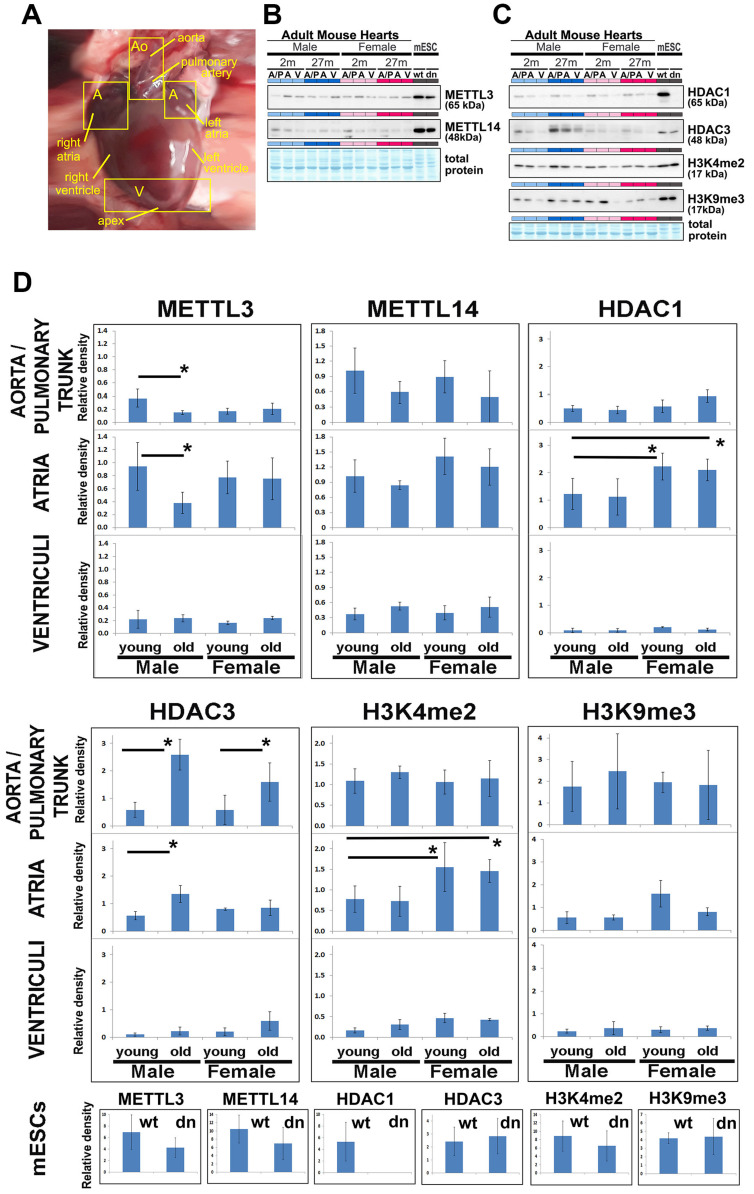
Epitranscriptomic and epigenetic data in mouse hearts, isolated from young and old animals. Data were compared with protein levels measured in HDAC1 wt and HDAC1 dn mESCs. (**A**) Adult mouse hearts were sectioned following the anatomy. Ventricular parts (V), atrium (A), or the aorta/pulmonary trunk (A/P). These anatomical regions were studied by Western blots showing the levels of (**B**) METTL3, METTL14 proteins, or (**C**) histone deacetylase 1 and histone deacetylase 3 (HDAC1, HDAC3), H3K4 di-methylation (H3K4me2), and H3K9 tri-methylation (H3K9me3). Data were quantified and related to the total protein levels; for quantification, see panel (**D**). The amido-black staining was used to show total protein levels that were measured by the µQuant^™^ spectrophotometer. Results originate from 3 biological replicates, and representative images were selected. Data are shown as the mean ± standard error of measurement (S.E.M.). Significantly different results (α = 0.05) are shown by asterisks. Lines indicate a comparison between two experimental groups. Data were analyzed by the Mann–Whitney U test.

**Figure 3 ijms-21-08139-f003:**
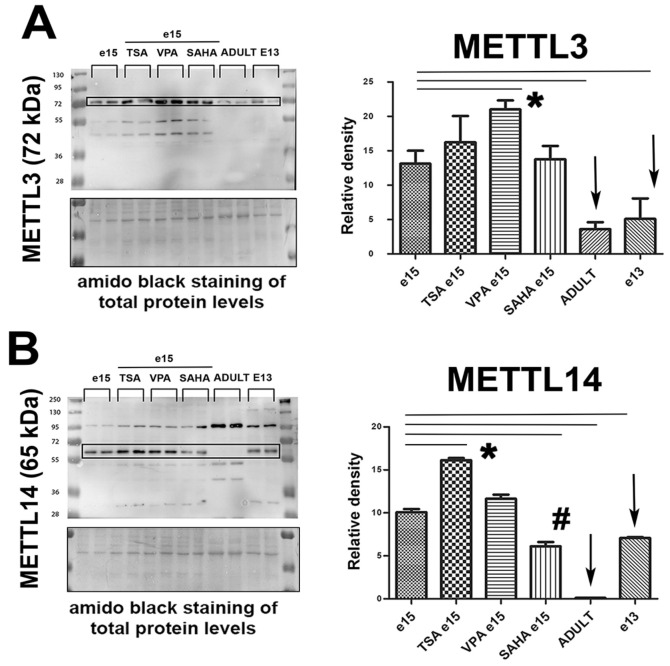
METTL3, METTL14, and METTL16 proteins studied in embryonic hearts treated with HDAC inhibitors. In non-treated, (Trichostatin A (TSA)-, valproic acid (VPA)-, and suberoylanilide hydroxamic acid (SAHA)-treated explanted mouse hearts at stage e15 following proteins we analyzed: (**A**) METTL3, (**B**) METTL14, and (**C**) METTL16. Western blot data were also compared with the levels of METTL-like proteins in hearts explanted from adult mice. (**D**) The effect of HDAC inhibitors was analyzed in e13, and e15 embryonic hearts treated with TSA, SAHA, and VPA and compared with adult hearts or mouse embryonic stem cells (mESCs). Western blot analyses show the levels of α-actinin and troponin. The density of Western blot fragments was analyzed by Image Studio Ver 5.2 software (LI-COR, Lincoln, NE, USA), and data were normalized to the total protein levels. Statistical analysis was performed by Student’s t-test (Sigma plot software, Systat Software, Inc, San Jose, CA, USA), and statistically significant differences (at *p* ≤ 0.05) from non-treated tissues are shown by asterisks (*). Results originate from 3 sets of Western blots; each contains 2 biological replicates (each replicate is a mixture of 3 embryonic hearts), as shown in panels (**A**–**D**). For statistical analysis in panel (**D**), the nonparametric Mann–Whitney U test was used (α = 0.05). Data are shown as the mean ± S.E.M.

**Figure 4 ijms-21-08139-f004:**
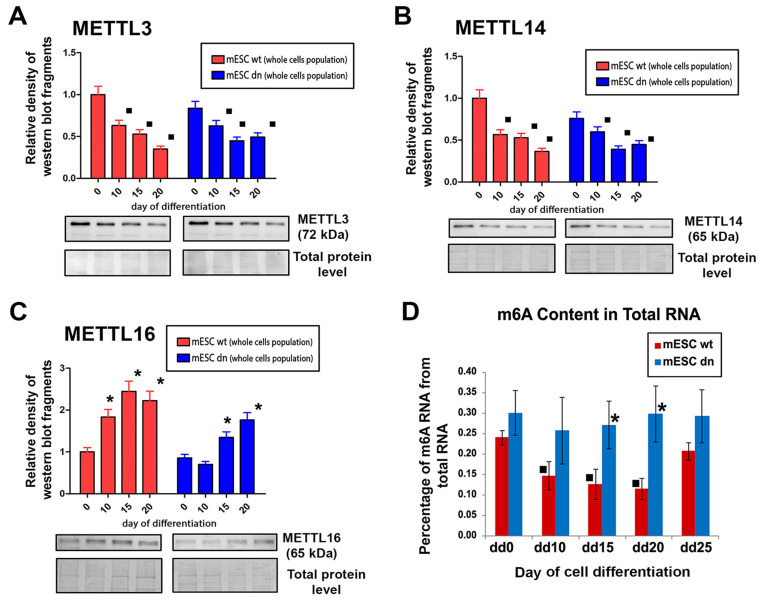
The levels of METTL3/METTL14, and METTL16 or m6A in RNAs analyzed in HDAC1 wild type (wt) and HDAC1 depleted (dn) mESCs differentiated into multicellular cell populations, including cardiomyocytes. Data from Western blots are shown in the following panels (**A**) METTL3, (**B**) METTL14, and (**C**) METTL16 in HDAC1 wt mESCs and HDAC1 dn mESCs. Squares (▪) show statistically significant differences at *p* ≤ 0.05 (a decrease in protein levels during differentiation). Data are significantly different from control values at dd0. Asterisks (*) show statistically significant differences (an increase) in protein levels. (**D**) Methylation of N^6^-adenosine in RNAs was analyzed by the use of colorimetry in non-differentiated and differentiated HDAC1 wt and HDAC1 dn cells at days 0, 10, 15, and 25 of differentiation. This cell population contained cardiomyocytes. The nonparametric Mann–Whitney U test was used for statistical analysis (α = 0.05). For all analyses, we used results from 3 biological replicates, each consisting of 3 technical replicates (panels (**A**–**C**)). Results in panel (**D**) originate from data of two biological replicates, each consisting of 3 technical replicates.

**Figure 5 ijms-21-08139-f005:**
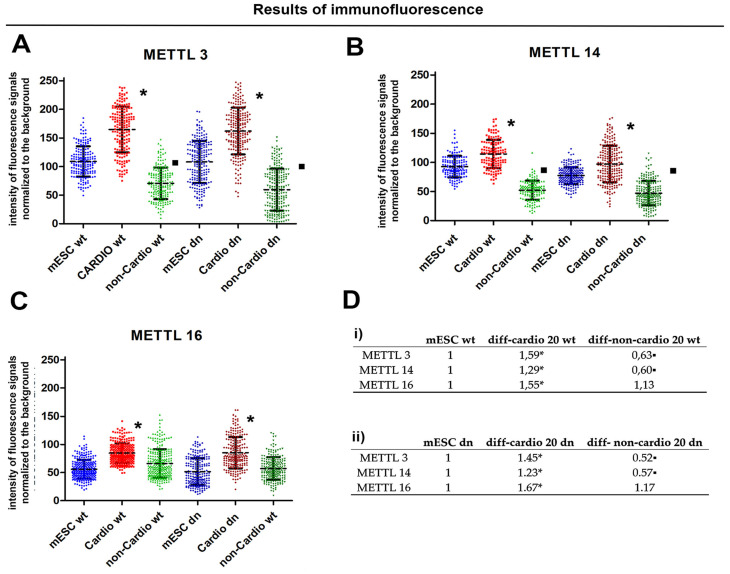
METTL-like proteins were analyzed in HDAC1 wt and HDAC1 dn mouse ESCs. (**A**) Fluorescence intensity of (**A**) METTL3, (**B**) METTL14, and (**C**) METTL16 in HDAC1 wt mESCs and HDAC1 dn mESCs. Asterisks (*) show increased levels of proteins studied during cell differentiation and compared with control non-differentiated cells. Squares (▪) show decreased levels of proteins. As statistically significant differences, marked by asterisk or squares, are considered results different from control value (at *p* ≤ 0.05). The Student’s t-test was used for statistical analysis. Experiments were repeated trice (panels (**A**–**C**)); 200–300 cells were analyzed for each experimental event. (**D**) The table shows numerical values (fold changes) linked to panels (**A**–**C**). Fold-change is an estimation of the data ratio (a ratio of the two means).

**Figure 6 ijms-21-08139-f006:**
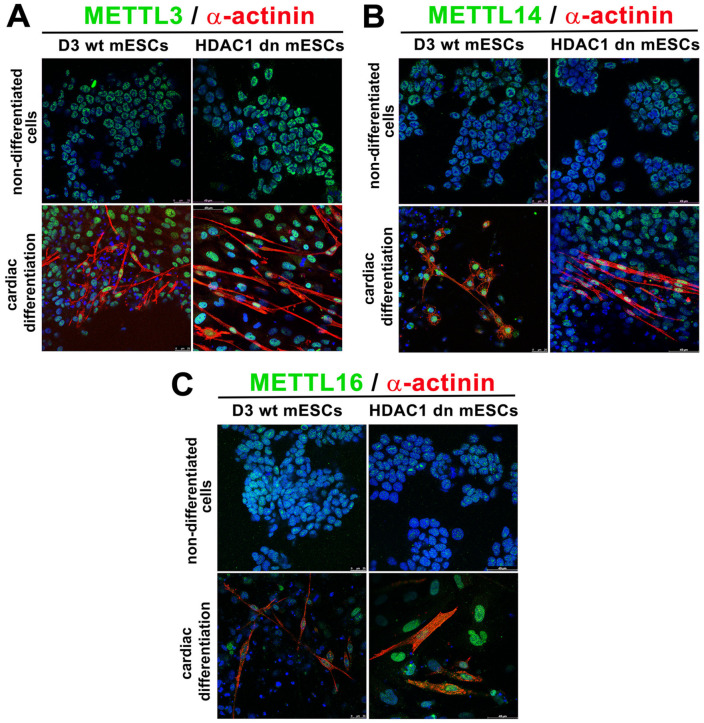
METTL3 and METTL14 proteins in α-actinin positive and α-actinin negative cells. Distribution profiles of (**A**) METTL3, (**B**) METTL14, and (**C**) METTL16 proteins (green) are shown in α-actinin positive (red) HDAC1 wt and HDAC1 dn mESCs. Other cell types were visualized only, according to 4’,6-diamidino-2-phenylindole (DAPI)-positivity (blue). Experiments were repeated twice, in each experiment, a 20–30 confocal scan with an identical cell density was performed.

**Figure 7 ijms-21-08139-f007:**
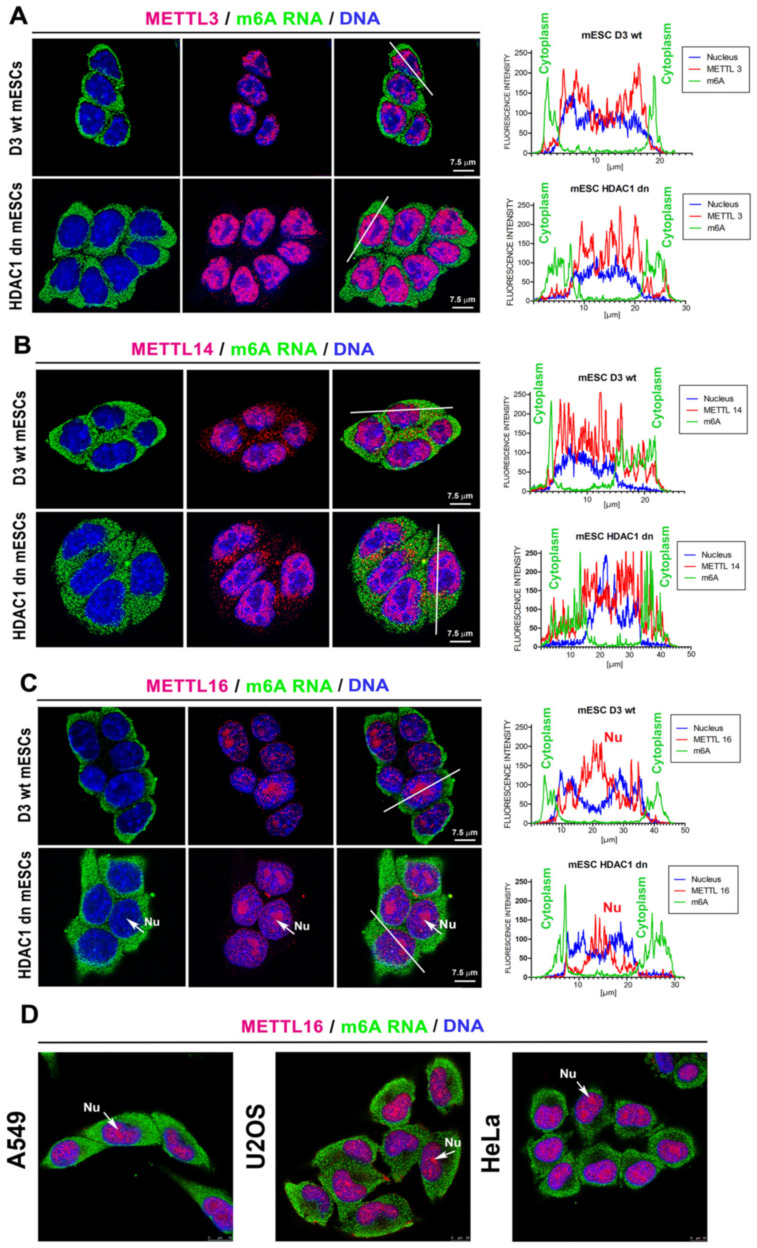
Localization of METTL-like proteins (red) and m6A RNAs (green) in HDAC1 wt and HDAC1 dn mESCs. The analysis was performed for (**A**) METTL3 (red), and m6A RNAs (green), (**B**) METTL14 (red) and m6A RNAs (green), or (**C**) METTL16 (red) and m6A RNAs (green). (**D**) For comparison with mESCs, the distribution pattern of the METTL16 protein was also analyzed in human tumor cells, A549, HeLa, and U2OS. Scale bars show 7.5 µm. Experiments were repeated twice; at least 200 cell nuclei were analyzed for each experimental event.

**Table 1 ijms-21-08139-t001:** Fold-Changes in the Levels of METTL-like Proteins in Distinct Anatomical Regions of e15 Embryonic Hearts.

	Aorta and Pulmonary Trunk	Right Atrium	Left Atrium	Intraventricular Septum	Right Ventricle	Left Ventricle
**METTL3**	1	1.30 *	1.16	0.80 ▪	1.26 *	1.09
**METTL14**	1	1.30 *	1.01	0.87 ▪	1.09	0.98
**METTL16**	1	3.76 *	2.44 *	1.34 *	1.74 *	1.60 *
**m6A RNA**	1	1.25 *	1.94 *	1.45 *	1.71 *	1.62 *

The ratio of fluorescence intensity (FI) in different regions of mouse embryonic hearts is shown. Data are normalized to the FI of the background and then compared to the FI in the aorta (value 1). The intensity of fluorescence was measured in Figure 1A(a–d). Fold-changes are shown as a ratio of two means. Asterisks (*) show increased level of proteins, squares (▪) show decreased protein levels.

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
