# Peer review of "The Distinct Function and Localization of METTL3/METTL14 and METTL16 Enzymes in Cardiomyocytes"

_ijms, 2020, doi:10.3390/ijms21218139_

Round 1
Reviewer 1 Report
The manuscript by Arcidiacono et al. shows that RNA methyltransferases METTL3/METTL14 and METTL16 have different distribution pattern in cardiomyocytes, mouse embryonic stem cells and in embryonic and adult mouse hearts. METTL3/METTL14 levels are distinct from the level of METTL16 and correlate better with the m6A RNA profile during differentiation. The authors conclude that METTL3/METTL14 and METTL16 have distinct functions in cardiomyogenesis.
Most of the data presented in the manuscript are of good quality, conclusions drawn are correct and based on solid data.
Major points.
- Distribution pattern of METTLs and m6A RNA in mESCs. In Fig.5, the images of all channels (colors) should be shown separately. Using merged images, it is difficult to follow fine patterns. In case of METTL14/m6A RNA (Fig.5B), some yellow dots are seen in the cytoplasm, does METTL14 show also some cytoplasmic localization?
- METTL16 localizes dominantly in the nucleolus of mESC cells (fig. 5C). For detection, HPA 020352 antibody (page 14, lane 398) was used. IF results with the same antibody are available on https://www.proteinatlas.org/ENSG00000127804-METTL16/cell, where this antibody detects signal in nucleoplasm and also cytosol using three different cancer cell lines. Does it mean that METTL16 has different localization in embryonic stem cells and transformed cell culture cell lines? Can you provide a control image with some cell culture cell line using the same antibody that in mESC cells? If this is a case, different localization may refer to different targets (rRNA?) in stem cells.
- Results section 2.2 – in mouse ESCs undergoing differentiation METTL3/14 levels are decreasing and METTL16 level increases. The same result is observed both in mESC wt and dn cells (Fig. 2 Bc). In Fig. 3, the change of m6A content in total RNA is different remaining unchanged in case of mESC dn cells and decreases in wt cells. In page 5, lane 178, the authors claim that “there should be a negative regulation of m6A in RNAs by METTL16”. My question is, if there is a negative regulation, why does it not take place in mESC dn cells?
- Figure 2 A. Fluorescence intensity (of what?). For example: page 5, lane 164, 166 – Western blot showed …. Fig. 2 Aa, b. Then on page 7, lanes 207, 208 – analysis of immunofluorescence data show…. Fig. 2Aa, Ab. So, what it is?
Figure 2 in its current form is confusing. I suggest to separate panels A and B and show them on different figures. The results of Fig. 2B are compared with m6A RNA levels showed in Fig. 3, they fit better together.
- In all figure legends, the number of analysis (measurements) used for statistics should be shown.
Minor points
- It was difficult to follow the manuscript because of a mess in referring to the figures in Results section. Too many typos.
Fig. 2 – Figure legend should be supplemented
Page 8, lane 212 – Fig. Ba, b ?
- Page 12, lanes 342-343 – Interestingly, in adult hearts, the METTL16 protein was under detection limit, in comparison to the METTL16 level in embryonic and adult hearts – repetition
Author Response
We thank the reviewers for his/her additional suggestions and interesting comments regarding how to improve our manuscript. Herein, we have replied to the reviewers’ criticisms. In the revised manuscript, changes are denoted with red fonts. Moreover, we would like to confirm that our manuscript was written according to the instructions of the IJMS journal, and we acknowledge that our work has not been previously published, that it is not under consideration for publication elsewhere, and that the text has been approved by all authors.
Reviewer 1
The manuscript by Arcidiacono et al. shows that RNA methyltransferases METTL3/METTL14 and METTL16 have different distribution patterns in cardiomyocytes, mouse embryonic stem cells, and in embryonic and adult mouse hearts. METTL3/METTL14 levels are distinct from the level of METTL16 and correlate better with the m6A RNA profile during differentiation. The authors conclude that METTL3/METTL14 and METTL16 have distinct functions in cardiomyogenesis.
Most of the data presented in the manuscript are of good quality, conclusions drawn are correct and based on solid data.
Major points.
- Distribution pattern of METTLs and m6A RNA in mESCs. In Fig.5, the images of all channels (colors) should be shown separately. Using merged images, it is difficult to follow fine patterns. In the case of METTL14/m6A RNA (Fig. 5B), some yellow dots are seen in the cytoplasm, does METTL14 show also some cytoplasmic localization?
Answer: In the revised version, we show RG panels and RGB overlay in Fig. 5. METTL14 signals are also in the cytoplasm. Also, we performed quantification of fluorescence in all cases studied, such analysis confirmed METTL14 in the cytoplasm and METTL16 in the cell nucleolus of mESCs.
- METTL16 localizes dominantly in the nucleolus of mESC cells (fig. 5C). For detection, HPA 020352 antibody (page 14, lane 398) was used. IF results with the same antibody are available on https://www.proteinatlas.org/ENSG00000127804-METTL16/cell, where this antibody detects a signal in the nucleoplasm and also cytosol using three different cancer cell lines. Does it mean that METTL16 has different localization in embryonic stem cells and transformed cell culture cell lines? Can you provide a control image with some cell culture cell line using the same antibody that in mESC cells? If this is a case, different localization may refer to different targets (rRNA?) in stem cells.
Answer: By immunohistochemistry, we studied the distribution pattern of METTL16 protein in tumor cells: A549, HeLa, and U2OS cells, and in all tested cell lines, METLL16 was dense inside the cell nucleoli (see revised Fig. D).
- Results section 2.2 – in mouse ESCs undergoing differentiation METTL3/14 levels are decreasing and METTL16 level increases. The same result is observed both in mESC wt and dn cells (Fig. 2 Bc). In Fig. 3, the change of m6A content in total RNA is different remaining unchanged in the case of mESC dn cells and decreases in wt cells. On page 5, lane 178, the authors claim that “there should be a negative regulation of m6A in RNAs by METTL16”. My question is if there is a negative regulation, why does it not take place in mESC dn cells?
Answer: The reviewer is completely right that this claim is wrong because, in the case of negative regulation of m6A in RNA by METLL16, there should be a decrease of m6A-RNA in bot HDAC1 wt and HDAC1 dn cells, while METTL16 was increased in both cell types studied. In the revised version, we deleted this wrong claim.
- Figure 2 A. Fluorescence intensity (of what?). For example, page 5, lane 164, 166 – Western blot showed …. Fig. 2 Aa, b. Then on page 7, lanes 207, 208 – analysis of immunofluorescence data show…. Fig. 2Aa, Ab. So, what it is?
Answer: In the case of western blots we quantified the intensity of fluorescence normalized to the background. In the case of western blots, we analyzed the density of western blots normalized to the total protein level.
For explanation, the levels of METTL3/14 were different when we analyzed the whole-cell population containing up to 30% of cardiomyocytes by western blots on one side, and another side alpha-actinin positive cardiomyocytes and non-cardio cells in the same cell population exposed to differentiation stimuli by immunofluorescence (IF). In this case, IF enables analysis on at an individual cell level, thus we were able to distinguish alpha-actinin positive and alpha-actinin negative cells.
Figure 2 in its current form is confusing. I suggest to separate panels A and B and show them on different figures. The results of Fig. 2B are compared with m6A RNA levels showed in Fig. 3, they fit better together.
Answer: In the revised version, Figs 2 and 3 were reorganized, as suggested by the reviewer.
- In all figure legends, the number of analyses (measurements) used for statistics should be shown.
Answer: A number of o analyses, samples, repetitions, and measurements we add to the legend of all images.
Minor points
- It was difficult to follow the manuscript because of a mess in referring to the figures in the Results section. Too many typos.
Answer: Typographical errors were corrected by a native speaker, and by professional Grammarly software.
Fig. 2 – Figure legend should be supplemented
Page 8, lane 212 – Fig. Ba, b?
Answer: This part was corrected in the revised version.
- Page 12, lanes 342-343 – Interestingly, in adult hearts, the METTL16 protein was under the detection limit, in comparison to the METTL16 level in embryonic and adult hearts – repetition
Answer: In the revised version, repetitions were eliminated, and mentioned part of the text was re-written.
Reviewer 2 Report
I read with interest the study by Arcidiacono et al. The subject of the study is very interesting and timely. Anyhow, the introduction is very long, and some concept can be move in discussion. Indeed, the discussion could be more structured in order to highlight the strength of the study and the significance of results. Also, the limitations of the study are totally missing. The authors should clearly draw their conclusions as well as indicate further steps to the knowledge in this field.
To understand the strength of the study and the significance of results the definition of n-values of hearts or cells regarding each type of experiment has to be reported. In method’s section, a paragraph describing the statistical analyses must be added.
Author Response
We thank the reviewers for his/her additional suggestions and interesting comments regarding how to improve our manuscript. Herein, we have replied to the reviewers’ criticisms. In the revised manuscript, changes are denoted with red fonts. Moreover, we would like to confirm that our manuscript was written according to the instructions of the IJMS journal, and we acknowledge that our work has not been previously published, that it is not under consideration for publication elsewhere, and that the text has been approved by all authors.
Reviewer 2
I read with interest the study by Arcidiacono et al. The subject of the study is very interesting and timely. Anyhow, the introduction is very long, and some concepts can be a move-in discussion.
Answer: Introduction was rewritten and some information from the Introduction sections was discussed in the revised Discussion.
Indeed, the discussion could be more structured in order to highlight the strength of the study and the significance of results.
The discussion was re-organized and we pointed out the main results.
Also, the limitations of the study are totally missing. The authors should clearly draw their conclusions as well as indicate further steps to the knowledge in this field.
Answer: Limitation of the study and future directions we mentioned in the Discussion section.
To understand the strength of the study and the significance of results the definition of n-values of hearts or cells regarding each type of experiment has to be reported. In the method’s section, a paragraph describing the statistical analyses must be added.
Statistical analyses were improved and the number of samples was mentioned in individual the figure legends. For data analysis, we used the Mann-Whitney U test (STATISTICA software).
Reviewer 3 Report
The manuscript entitled “The distinct function and localization of METLL3/METTL14 and METTL16 enzymes in cardiomyocytes” by Arcidiacono et. analyzed the distribution of METTL3/METTL14 (methyltransferase-like 3/methyltransferase-like 14), METTL16 (methyltransferase-like 16) and N6-methyladenosine (m6A) during cardiomyogenesis in wild type (wt) and HDAC1 (histone deacetylase 1) double negative (dn) mouse embryonic stem cells (mESCs) as well as in young and old mouse hearts. Using immunofluorescence and western blots they found different expression pattern of these writers of m6A where METTL3/METTL14 and METTL16 differ.
Comments:
- This is an interesting investigation. Unfortunately the result part is a bit difficult to read. This part should be better structured. Some sentences cannot be assigned and are lost in the text. Some of the images are of poor quality (lack of resolution), so that the results are not easy to judge. It is unfavorable to switch between total hearts and embryonic cells. It may be more convenient to start with adult hearts first, and then move on to embryonic hearts, then the mESCs and finally the cells with the inhibitors.
- There is no separate statistics section. In general, the statistics are much too limited. The number of examined hearts is not mentioned.
- Figure 1: resolution too low. Immunofluorescence. Whereupon the immunofluorescence is normalized? Since the authors present box plots, several hearts must have been evaluated. What is the number of hearts? Are there statistically differences?
- What is the difference between figure 1B and table 1? Why are the data normalized to the aorta?
- Figure 2: Are these the individual ESCs in the graph for fluorescence? How many cell do you count and on how many dishes or how many individual bottles were they cultured? The normalization of the western blot is missing.
- Table 2: Should be shown together with Figure 3 or integrated into the figure
- Figure 3: What do the percentages refer to? N=?
- Figure 6 is too small. In addition, resolution must be increased. Please show a quantification of female and male expression and for WT and HDAC1 dn cells. N=?
- Figure 7 is too small. In addition, resolution must be increased.
- Why do the author use day 20 of cardiomyogenesis for α-actinin-positive cells and for the remaining experiments day 15?
- Since HDAC inhibitors have effects on hypertrophy, the author could use some hypertrophy markers to substantiate their results.
- The authors sometimes write HDAC1 double knockouts. This is misleading, because you think of different isoform; better HDAC1 double negative or HDAC1 depleted cells.
Author Response
We thank the reviewers for his/her additional suggestions and interesting comments regarding how to improve our manuscript. Herein, we have replied to the reviewers’ criticisms. In the revised manuscript, changes are denoted with red fonts. Moreover, we would like to confirm that our manuscript was written according to the instructions of the IJMS journal, and we acknowledge that our work has not been previously published, that it is not under consideration for publication elsewhere, and that the text has been approved by all authors.
Reviewer 3
- This is an interesting investigation. Unfortunately, the result part is a bit difficult to read. This part should be better structured. Some sentences cannot be assigned and are lost in the text. Some of the images are of poor quality (lack of resolution) so that the results are not easy to judge. It is unfavorable to switch between total hearts and embryonic cells. It may be more convenient to start with adult hearts first, and then move on to embryonic hearts, then the mESCs, and finally the cells with the inhibitors.
Answer: In the revised version, the result part was improved. Images are poor quality because it is pdf, but we can provide images in tiff format.
Answer: It is a good idea to start with adult hearts and continue with embryonic ones, and finally with mESCs.
- There is no separate statistics section. In general, the statistics are much too limited. The number of examined hearts is not mentioned.
Answer: Statistics were improved and a number of samples and the number of biological replicates we mentioned in the figure legends.
- Figure 1: resolution too low. Immunofluorescence. Whereupon the immunofluorescence is normalized? Since the author's present box plots, several hearts must have been evaluated. What is the number of hearts? Are there statistical differences?
Answer: Images are poor quality because the manuscript is in pdf format – see submitted Word format. We can also provide images in tiff format.
For statistic, we used the Mann Whitney test from STATISTICA software or the following web site: https://www.socscistatistics.com/tests/mannwhitney/default2.aspx
The number of hearts is mentioned in the figure legend, but we did not evaluate fluorescence among heart, just only among selected anatomical regions in identical heart sections.
- What is the difference between figure 1B and table 1? Why are the data normalized to the aorta?
Answer: In graphs 1B, OA compared the distribution profile of METTLs inside different parts of the heart among different heart sections. So, it was necessary to define a common “internal CTRL” among the sections to compare the distribution profiles. OA chooses the aorta because it is a well visible and quite constant structure among all the heart sections.
Table 1 shows the numerical fold-change of fluorescent intensity using as control the Aorta measurement (value equal to 1). All the numerical value is the average of the populations of measured values for each part, then divided by Aorta value.
- Figure 2: Are these the individual ESCs in the graph for fluorescence? How many cells do you count and on how many dishes or how many individual bottles were they cultured? The normalization of the western blot is missing.
Answer: we have indicated the number of cells in the methodological part see:
“Immunofluorescence on Mouse Embryonic Cells and Heart Sections and Confocal Microscopy”…” We monitored 200-300 cell nuclei and 20 cryosections, and we performed an analysis of 2-3 biological replicates. The images were acquired in an 8-bit setting, corresponding to a display range of 256 gray levels. LEICA LAS AF software was used for image acquisition and analysis of fluorescence intensity“.
The WB data were normalized to the total protein level.
- Table 2: Should be shown together with Figure 3 or integrated into the figure
Answer: In the revised version, Table 2a, 2b was integrated into the revised version of Fig. 3.
Figure 3: What do the percentages refer to? N=?
Answer: For results in panel revised Fig. 3D we used data from two biological replicates, each consisting of 3 technical replicates.
- Figure 6 is too small. In addition, the resolution must be increased. Please show the quantification of female and male expression and for WT and HDAC1 dn cells. N=?
Answer: Revised Fig. 6 was enlarged, we increased resolution, and data from western blots were quantified by ImageJ software. We pointed out the results showing differences between mESCs wt and mESCs HDAC1 dn in which we studied the following proteins: METTL3, METTL14, HDAC1, HDAC3, H3K4me2, and H3K9me3.
- Figure 7 is too small. In addition, the resolution must be increased.
Answer: the size of the image was enlarged and we increased the resolution of this figure. Also, we can provide a tiff file for the final version if demanded.
- Why does the author use day 20 of cardiomyogenesis for α-actinin-positive cells and for the remaining experiments day 15?
Answer: In our experiments, we have always used until the 20th day of differentiation. We don’t see this error in the text.
- Since HDAC inhibitors have effects on hypertrophy, the author could use some hypertrophy markers to substantiate their results.
Answer: In samples used for Fig. 3D we additionally analyzed troponin (ab47003, Abcam), as a marker of hypertrophy. Siciliano et al. (2000) showed that troponin I is a marker of myocardial damage, higher serum values probably indicate a more important cardiac involvement in the setting of hypertrophy disease. In revised Fig. 3D we showed that the level of troponin is increased in E15 hearts treated by VPA and is very high in adult hearts – see result part and the discussion section.
- The authors sometimes write HDAC1 double knockouts. This is misleading because you think of different isoform; better HDAC1 double negative or HDAC1 depleted cells.
Answer: We used the term HDAC1 depleted cells
Round 2
Reviewer 1 Report
Thank you for rearrangement you made in the manuscript. It is much easier to follow it now.
Author Response
Many thanks to the reviewer for the revision of our manuscript and useful recommendations.
All the best,
Eva Bartova
Reviewer 3 Report
The manuscript entitled “The distinct function and localization of METLL3/METTL14 and METTL16 enzymes in cardiomyocytes” by Arcidiacono et. analyzed the distribution of METTL3/METTL14 (methyltransferase-like 3/methyltransferase-like 14), METTL16 (methyltransferase-like 16) and N6-methyladenosine (m6A) during cardiomyogenesis in wild type (wt) and HDAC1 (histone deacetylase 1) double negative (dn) mouse embryonic stem cells (mESCs) as well as in young and old mouse hearts. Using immunofluorescence and western blots they found different expression pattern of these writers of m6A where METTL3/METTL14 and METTL16 differ. The authors have extensively revised the manuscript and added missing information. In addition, the resolution of the pictures is better. Nevertheless, there are still some points that need to be changed.
Comments:
- In figure 1 A the authors show the distribution of METTL3/14/16 protein and m6A RNA in the embryonic heart and in 1B the quantification of the measurement. They show box plots with minimum, maximum and median I suppose. Since they only analyzed two hearts (“experiments were repeated twice”) the analyses must have been performed with n=40 (2x 20 cryo-sections). Therefore, the statistic analysis can also be displayed in the box plots. In comparison to aorta, METTL16 is statistically higher expressed in all other areas. In left atrium, (LA) METTL16 is also two times higher expressed. Looking at the deviations of LA and pulmonary trunk (PT), the increased expression is more stable in LA than PT. The heading would then have to be adjusted here.
- The figure legends are not complete in 1Bc and 1Bd.
- In table 1 the authors present the quantification by referring all other regions to the aorta. Please show not only the mean but also the standard deviation. Are there also quantitative differences between the different areas? The data are normalized to the background. Then, they compare the areas to the background of the aorta, why? They have to compare it to the aorta normalized to the background!
- Why do the authors name aorta + pulmonary trunk aorta-associated vessels?
- Figure 2 shows western blot of different proteins from young and adult mice. In the figure, the old mice are 27 month old; in the text, they are 28 month old.
- In the figure legend 2 it is missing whether the authors show mean ±SE or SEM.
- The y-axis is always different in the graphs. This makes a direct comparison difficult. Besides, the label of the axis is difficult to read.
- Figure2: METTL3 is detectable in aorta+pulmonary trunk (AP), atria and ventricle. From the “quantification”, the density of METTL3 in aorta is comparable to the ventricle. If you look at the standard deviations, the differences are also very high. Figure 2 shows no statistic evaluation at all.
- The author state that TSA and VPA increases the levels of all proteins. Since the authors only used two biological replicates it's hard to claim that. Also due to the artificial increase by applying the samples three times in the WB, only VPA for METTL3 and TSA for METTL14 is increased. The sample size must be increased.
- In the figure legend 3 it is missing whether the authors show mean ±SE or SEM.
- The signs of significance have slipped into the images in figure 3.
- In the figure legend it is missing whether the authors show mean ±SE or SEM. Do the author carry out statistics with n = 3 or n = 9?
- In figure 5 A-C the fluorescence intensity is shown. Howe many cells did the author count? In 5D, please add the standard deviation.
- In figure 5 the experiments were done trice. Does this mean that the ESC come from three different preparations/approaches?
- Figure 6 show α-actinin positive cells. Is there a missing inscription on the left side of the Pictures?
- Statistic: As different areas are compared with each other, a group analysis should be carried out – ANOVA.
Author Response
We thank the reviewer for his/her additional suggestions and comments. We have replied to the reviewer's criticisms. In the revised manuscript (the 2nd revision), changes are denoted with green fonts. Our responses, see below:
The manuscript entitled "The distinct function and localization of METLL3/METTL14 and METTL16 enzymes in cardiomyocytes" by Arcidiacono et. analyzed the distribution of METTL3/METTL14 (methyltransferase-like 3/methyltransferase-like 14), METTL16 (methyltransferase-like 16) and N6-methyladenosine (m6A) during cardiomyogenesis in wild type (wt) and HDAC1 (histone deacetylase 1) double-negative (dn) mouse embryonic stem cells (mESCs) as well as in young and old mouse hearts. Using immunofluorescence and western blots they found different expression pattern of these writers of m6A where METTL3/METTL14 and METTL16 differ. The authors have extensively revised the manuscript and added missing information. In addition, the resolution of the pictures is better. Nevertheless, there are still some points that need to be changed.
Comments:
- In figure 1 A the authors show the distribution of METTL3/14/16 protein and m6A RNA in the embryonic heart and in 1B the quantification of the measurement. They show box plots with minimum, maximum and median I suppose. Since they only analyzed two hearts ("experiments were repeated twice") the analyses must have been performed with n=40 (2x 20 cryo-sections). Therefore, the statistic analysis can also be displayed in the box plots. In comparison to aorta, METTL16 is statistically higher expressed in all other areas. In left atrium, (LA) METTL16 is also two times higher expressed. Looking at the deviations of LA and pulmonary trunk (PT), the increased expression is more stable in LA than PT. The heading would then have to be adjusted here.
Answer: These graphs (Fig. 1Ba-d) are valuable to show the distribution pattern of the METTLs. According to what the reviewer suggested, we removed the PT measurement from the graph (also considering that PT measurement is present just in METTL 16 and m6A graphs); we changed the heading.
To our knowledge, many authors are using protein level detection by immunofluorescence (IF); thus, we also show the same, but we are aware that western blots (WB) provide better quantification, as shown in Fig. 2A-D. By WB, it is possible to study adult hearts, but sectioning of e15 embryonic hearts is really impossible.
- The figure legends are not complete in 1Bc and 1Bd.
Answer: This part of the Figure was revised, and the figure legend was improved
- In table 1 the authors present the quantification by referring all other regions to the aorta. Please show not only the mean but also the standard deviation. Are there also quantitative differences between the different areas? The data are normalized to the background. Then, they compare the areas to the background of the aorta, why? They have to compare it to the aorta normalized to the background!
Answer: Table 1 is a table showing the fold-change. The SD is a property of the data, but fold-change is an estimation of the data (basically a ratio of the two means), so it is not possible to show SD.
We do not compare the areas in the background of the aorta. In each individual measurement, we subtracted the fluorescence (FI) of the whole background in the given channel from FI acquired for the given protein of interest. Then, Orazio Arcidiacono compared all adjusted measurements to the adjusted value of the aorta. So, basically, the corrected fluorescence values (subtraction of fluorescence background) were normalized to corrected aorta values.
- Why do the authors name aorta + pulmonary trunk aorta-associated vessels?
Answer: It is hard to be sure that we are able to take exactly individual aortas, so we wanted to express that it is the anatomical region around aortas – exactly, we took aortas/pulmonary trunks.
- Figure 2 shows western blot of different proteins from young and adult mice. In the Figure, the old mice are 27 month old; in the text, they are 28 month old.
Answer: We are sorry for this misprint – mice were 27 months old.
- In the figure legend 2 it is missing whether the authors show mean ±SE or SEM.
Answer: For individual experimental events, we showed the mean ± S.E.M. (standard error of measurement). This part of the figure legend was improved. We increased fonts, reorganized panel D, and we show our statistical analysis.
- The y-axis is always different in the graphs. This makes a direct comparison difficult. Besides, the label of the axis is difficult to read.
Answer: In a frame of one type of protein, the scale of axes is identical, but it is not possible to do this for all proteins due to differences in the binding of antibodies (see the distinct density of western blot fragments). The band density for ESCs was not possible to compare with METTL-related WB bands in tissue. It was due to a high level of proteins studied in ESCs; thus, the quantification of ESCs is shown separately. The labeling of axes was enlarged, and panel D was reorganized.
- Figure 2: METTL3 is detectable in aorta+pulmonary trunk (AP), atria and ventricle. From the "quantification", the density of METTL3 in aorta is comparable to the ventricle. If you look at the standard deviations, the differences are also very high. Figure 2 shows no statistic evaluation at all.
Answer: Statistics in Fig. 2D was improved. Significantly different results alpha =0.05 is shown by asterisks, and lines show what we compared.
For statistical analysis, we used the Mann-Whitney U test (STATISTICA software), which is a nonparametric test of the null hypothesis that is applied for X and Y values, randomly selected from two experimental units. The primary step of this analysis contains the U statistic involving the use of asymmetric real-valued function h(x,y). The following formula describes the approach:
Note: Both equations are in pdf file
- (n 2) is the binomial coefficient,
- (ut) are independent and identically distributed variables,
- Σ = summation notation
The following two formulas are applicable for the Mann-Whitney U Test. R is the sum of ranks in the sample, and n is the number of items in the sample.
We ran the test at the 5% level of significance (i.e., * means α=0.05).
As recommended by the review, the following observation we revised: the density of METTL3 in the aorta is comparable with the ventricle. We also, and more importantly, observed identical trends between aortas and atria, showing METTL3 down-regulation caused by aging in male animals (see asterisks in revised Fig. 2D).
The author state that TSA and VPA increases the levels of all proteins. Since the authors only used two biological replicates it's hard to claim that. Also due to the artificial increase by applying the samples three times in the WB, only VPA for METTL3 and TSA for METTL14 is increased. The sample size must be increased.
Answer: Result originates from 3 sets of western blots; each originates from 2 biological replicates, as shown in Fig. 3A-D. Each replicate is a mixture of 3 embryonic hearts. This mixture is essential to do in order to have a sufficient amount of proteins. A new verification of these results is enclosed in Figure for the reviewer. From this repeated WB we can conclude that HDACi, especially VPA up-regulates METLL-line proteins.
- In the figure legend 3 it is missing whether the authors show mean ±SE or SEM.
Answer: We show the mean ± S.E.M.
- The signs of significance have slipped into the images in figure 3.
Answer: We enlarged signs of significance. We tried to improve the labeling of individual panels. Fig. 3 was revised.
- In the figure legend it is missing whether the authors show mean ±SE or SEM. Do the author carry out statistics with n = 3 or n = 9?
Answer: We show mean ± S.E.M. Result originates from 3 sets of western blots; each originates from 2 biological replicates (each replicate is a mixture of 3 embryonic hearts – due to sufficient amount of proteins, essentially to be used for western blots – for technical repetitions of each protein of interest)
- In figure 5 A-C the fluorescence intensity is shown. How many cells did the author count? In 5D, please add the standard deviation.
Answer: 200-300 was evaluated.
The SD is a property of the data, but fold-change is an estimation of the data ratio (basically a ratio of the two means), so the SD makes no sense in this case. Anyway, standard errors of the measurement are well visible in the graphs (Fig 5A-C).
- In figure 5 the experiments were done trice. Does this mean that the ESC come from three different preparations/approaches?
Answer: We used three biological replicates, so it means that ESCs come from three different runs of experiments. ESCs were differentiated trice, and for analysis, we additionally used three technical replicates. It means that three slides for each experimental event were used for scanning; thus, we approximately analyzed 200-250 cells, as we mentioned in the Material and methods section.
- Figure 6 show α-actinin positive cells. Is there a missing inscription on the left side of the Pictures?
Answer: Green fluorescence means METTL proteins, and red fluorescence means alpha-actinin. Non-differentiated cells are alpha-actinin negative, and differentiated cells are alpha-actinin-positive. In the revised version, we added a description like non-differentiated cells and cardiac-differentiation. It is enclosed to the left side of panel A-C (Fig. 6).
- Statistic: As different areas are compared with each other; a group analysis should be carried out – ANOVA.
Answer: For statistical analysis, we newly used the Mann-Whitney U test (STATISTICA software), which is a nonparametric test that enables us to compare two independent samples when the outcome is not normally distributed, and the number of samples is small. We guess that this test is really suitable for our analysis. Many thanks to the reviewer for his/her suggestion. Instead of the ANOVA test, we decided to use the Mann-Whitney U test.

Round 3
Reviewer 3 Report
Minor comments:
- In the methods section the following antibodies used in the WB are missing α-actinin, troponin I.
- The definition for abbreviation “ao” is obviously wrong.
- Figure 7 is too small. This illustration should cover the entire width of the page. The legend of the figure can be placed under it.
- Please check the hyphens, for example: α-actinin-positive and α-actinin positive
- Please check tempi: change between present tense and past tense
- very small spelling mistakes, missing d (experimentally-induce differentiation), missing preposition, missing spaces
Author Response
Many thanks for mentioning minor points that we improved in the revised text. Changes are shown by violet fonts.
- In the methods section the following antibodies used in the WB are missing α-actinin, troponin I.
Answer: Antibody specification was used in the WB section: we used anti-α-actinin (A7811; Sigma-Aldrich, CZ), and anti-troponin (ab47003, Abcam, UK).
- The definition for abbreviation “ao” is obviously wrong.
Answer: we are aware that we have analyzed aorta and pulmonary trunk, so in the revised version, we replace AO with A/P in Fig. 2, but in Fig. 1 we were able to measure fluorescence intensity directly in the aorta, labeled as AO.
- Figure 7 is too small. This illustration should cover the entire width of the page. The legend of the figure can be placed under it.
Answer: In the revised version, we enlarged this figure as much as it was possible.
- Please check the hyphens, for example, α-actinin-positive and α-actinin positive
Answer: This part was uniformed, we used the term α-actinin positive.
- Please check tempi: change between present tense and past tense
Answer: Final version was verified by Grammarly software
- very small spelling mistakes, missing d (experimentally-induce differentiation), missing preposition, missing spaces
Answer: Final version was verified by Grammarly software
